# Preparation and Application of Carbon-Based Materials in the Production of Medium-Chain Carboxylic Acids by Anaerobic Digestion: A Review

**Lihua Jiao** [1,2,†], **Yang Liu** [2,†], **Chunhui Cao** [1,2], **Quan Bu** [1], **Mingqing Liu** [3,*] and **Yonglan Xi** [1,2,*]

1 School of Agricultural Engineering, Jiangsu University, Zhenjiang 212013, China; 2222216050@stmail.ujs.edu.cn (L.J.); 2212116034@stmail.ujs.edu.cn (C.C.); qbu@ujs.edu.cn (Q.B.)
2 Jiangsu Academy of Agriculture Sciences, Nanjing 210014, China; 202262118004@njtech.edu.cn
3 Nanjing Institute of Environmental Science, Ministry of Ecology and Environment, Nanjing 210042, China
* Correspondence: lmq@nies.org (M.L.); yonglanxi@jaas.ac.cn (Y.X.)
† These authors contribute equally to this work.

**Abstract:** The main purpose of this article is to explore the mechanism of action of carbon-based materials in the anaerobic digestion (AD) production of medium-chain carboxylic acids (MCCA). Currently, there are various methods to increase production, but there is no review on how carbon-based materials improve MCCA. This paper first introduced the chain elongation (CE) technology, focusing on the factors affecting the production of MCCA by AD, such as pH, temperature, the ratio of electron donor (ED) to an electron acceptor (EA), substrate type, and other related factors. This article introduces the preparation and characteristics of carbon-based materials, as well as the effect and mechanism of adding carbon-based materials to AD acid production. Finally, the shortcomings of the current research were pointed out, and future research directions were prospected, aiming to provide a reference for improving the efficiency of AD of MCCA using carbon-based materials.

**Keywords:** organic waste; carbon-based materials; chain elongation; medium-chain carboxylic acids



## 1. Introduction

Organic waste is increasing with human activities, and, if not treated promptly, it can have a significant impact on the environment [1]. How to handle organic waste has become a recent hot topic. According to statistics, global solid waste production in 2020 was 2.24 billion tons/year, and it is expected to reach 3.88 billion tons/year by 2050, with organic waste accounting for approximately 46% [2]. Converting this waste biomass into high-value chemicals is an important way to achieve the goals of "carbon peaking" and "carbon neutrality", and it is also regarded as an important way to develop the carbon cycle bio-based economy [3]. The current mainstream method is to convert organic waste into various valuable substances such as methane, $H_2$, short-chain carboxylic acids (SCCA), and MCCA through AD [4–6]. Among them, methane and $H_2$ can be used as fuels to alleviate greenhouse gas emissions, while SCCA is commonly used in a wide range of fields, including animal feed additives and biofuel production precursors [7]. However, MCCA has low solubility in water, is easy to extract, and has high economic value [8]. According to reports, the price of refined caproate is between $2000 and $3000 per ton, with a market size of 25,000 tons per year [9]. Therefore, chain elongation (CE) technology is usually used to extend the carbon chain of SCCA to MCCA [10]. MCCA is a monocarboxylate of organic acids with a carbon atom number of 6–12, used in a wide range of fields, including important chemical components of synthetic fragrances, surfactants, adhesives, lubricants, and plasticizers, as well as precursors for fuel production [11]. CE technology involves adding two carbon atoms to the carbon chain of SCCA each time, and repeating this process can convert SCCA into MCCA [12].

However, there are many inhibitory factors in the CE process, for example; when the substrate source is complex, the hydrolysis of the substrate is a rate-limiting step that can affect the production of MCCA [13]. At the same time, ammonia, sulfides, heavy metals, etc. in the substrate can also affect the micro-organisms of CE [14]. As the CE process continues, the concentration of undissociated carboxylic acids (undissociated hexanoic acid and octanoic acid) produced by it will inhibit the micro-organisms involved in the CE process [15]. Relevant studies have reported that when the undissociated carboxylic acid in the AD system reaches $0.2 \text{ g·L}^{-1}$, micro-organisms will be severely inhibited, and when the concentration of undissociated hexanoic acid reaches $0.8 \text{ g·L}^{-1}$, this is the limit for micro-organisms [16]. The concentration of the substrate (ethanol) can also have a toxic effect on micro-organisms. When the ethanol concentration reaches 200–400 mM, micro-organisms are also inhibited, leading to an impact on the generation of the final products [17]. The usual solution to the above problems is to reduce the concentration of inhibitory substances or enhance the resistance of micro-organisms to inhibitors—for example, the pre-treatment of substrates to eliminate the impact of hydrolysis (thermal hydrolysis process) [13]; the optimization of SCCA and MCCA processes in two stages of fermentation to reduce the environmental impact on MCCA [17]; the co-fermentation to obtain favorable C/N ratios [18]; and providing a buffering capacity and abundant carbon water compounds [19], and additives (including carbon and iron-based materials) to strengthen microbial communities [20,21]. pH regulation, the dilution of raw materials, and the online extraction of carboxylic acids eliminate the toxic effects of undissolved carboxylic acids on micro-organisms, etc. [9]. Adding carbon-based materials can improve direct interspecies electron transfer (DIET) in cells based on their conductivity and redox properties [22]. There is literature indicating that DIEF can be more thermodynamically advantageous, with less energy loss and a higher electron transfer rate in kinetics [23]. Carbon-based materials can form stable structures with micro-organisms through adsorption, alleviate the toxicity of micro-organisms to undissociated carboxylic acids and ethanol, enhance the stability of the CE system, shorten the lag time, and, thus, improve the production efficiency of MCCA [24–27]. At the same time, carbon-based materials can also be obtained from organic waste or AD residues, which can be used as a method for treating organic waste [28]. Currently, the added carbon-based materials include porous carbon materials and graphene. Therefore, a comprehensive understanding of the preparation methods and characteristics of different carbon-based materials is necessary and meaningful for studying the production of chain carboxylic acids in waste fermentation.

At present, there is no relevant literature review on the mechanism of carbon-based materials producing MCCA in AD. This paper mainly introduces the mechanism of organic waste acid production and focused on the influence factors of AD acid production, such as pH, temperature, substrate, the ratio of ED to EA, substrate type, and other factors. This article introduces the preparation and characteristics of carbon-based materials, as well as the application and mechanism of adding carbon-based materials in AD-produced MCCA. Finally, the shortcomings of the current research were pointed out, and future research directions were prospected, aiming to provide a reference for improving the efficiency of AD-produced MCCA using carbon-based materials.

## 2. Chain Elongation Technology

CE technology mainly includes the reverse β-oxidation (RBO) pathway and fatty acid biosynthesis (FAB) pathway [12]. The RBO pathway is a typical pathway widely used in CE technology [16]. The RBO pathway is centered around acetyl CoA, and two carbon atoms can be added to the starting molecule through each cycle. Repeating this process can produce MCCA compounds with 6–12 carbon atoms [29]. However, this process usually uses ED and EA as substrates, and micro-organisms eventually form a series of products through different types of metabolic pathways and enzyme catalysis [30]. At present, ethanol and lactic acid are mainly used as ED, while volatile fatty acids (VFA) such as acetic



acid, butyric acid, and propionic acid are used as EA [31]. The following mainly discusses CE technology driven by ethanol and lactic acid, as shown in Figure 1.

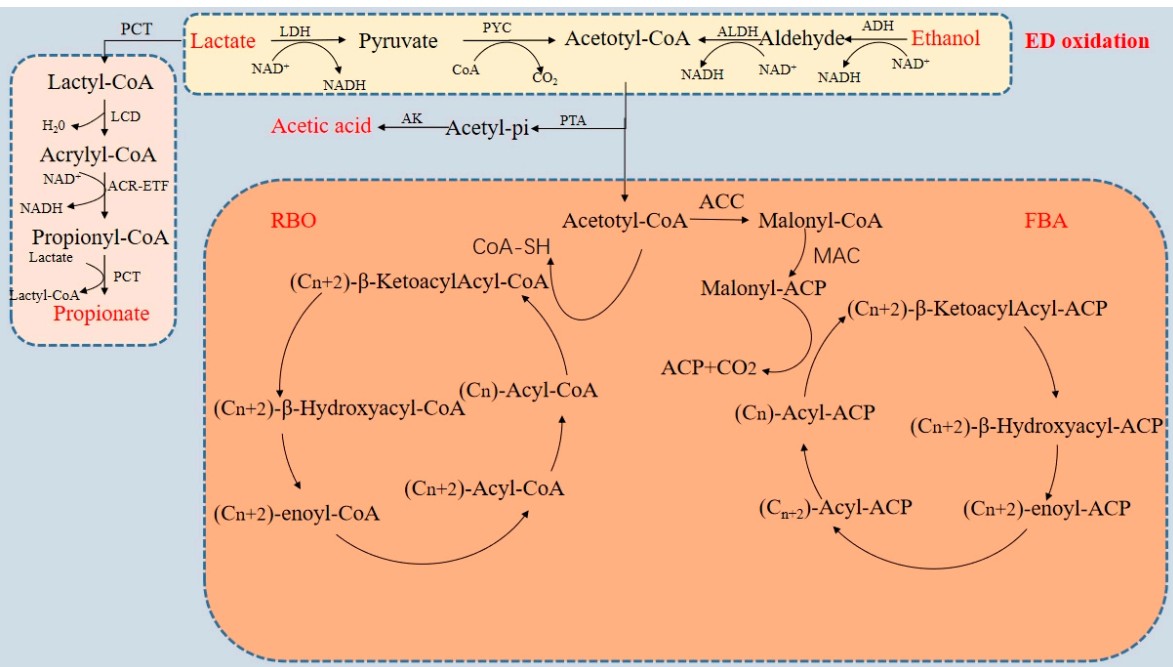

**Figure 1.** Schematic diagram of chain extension (according to Kim et al. (2022) [32], Wu et al. (2022) [31], and Xiang et al. (2022) [16]): ALDH, acetaldehyde dehydrogenase; LDH, lactate dehydrogenase; PYC, pyruvate carboxylase; PCT, propionyl-CoA transferase; LCD, lactyl-CoA dehydratase; AK, acetate kinase; PTA, phosphate acetyltransferase; ACC, acetyl-CoA carboxylase; and MAT, malonyltransfera.

In the RBO pathway, under anaerobic conditions, lactic acid is first converted to pyruvate through the catalysis of lactate dehydrogenase, and then acetyl CoA is generated through the action of pyruvate decarboxylase and acetaldehyde dehydrogenase [32]. However, not all acetyl CoA produced in this pathway will be converted to methyl propionyl CoA, as some acetyl CoA is oxidized to acetic acid to provide energy for micro-organisms (reaction formula as shown in Equation (1)) [9]. CE reactions are obtained via ethanol and lactic acid, and thermodynamic information with concentrations and pressures of all components at 1 M or 1 bar, pH 7 at 25 °C. Two acetyl CoA enzymes undergo three separate reactions to produce butyryl CoA, which is then transferred to acetic acid to form butyric acid (reaction formula as shown in Equation (2)) [33]. Similarly, butyryl CoA and acetyl CoA undergo a three-step reaction to produce hexyl CoA and octyl CoA, followed by the production of caproic acid and octanoic acid, respectively [12,32,34]. Among them, the reaction formula as shown in Equation (3) is the reaction formula for converting lactic acid to caproic acid [35]. It is worth noting that, as the concentration of lactic acid increases, the propionic acid pathway is enhanced, which converts lactic acid into propionic acid and affects the generation of MCCA [36]. Some micro-organisms with genes involved in the propionic acid pathway can also undergo reverse-transcription β-identification by oxidizing lactic acid to propionic acid through fewer oxidation pathways in the reaction steps [37]. The oxidation process of ethanol is different from that of lactic acid, as it undergoes the stage of ethanol oxidation to produce acetaldehyde [7]. Ethanol and acetaldehyde are catalyzed by ethanol dehydrogenase and acetaldehyde dehydrogenase, respectively, to produce acetyl CoA [38]. During this process, 1/6 of acetyl CoA is converted to acetic acid through substrate phosphorylation, while the remaining 5/6 enters the reverse-cycle β-oxidation pathway (reaction formula as shown in Equation (4)) [39]. Equations (5) and (6)

show the reaction of ethanol with acetic acid and its derivative Acetyl-CoA to produce caproic acid [9].

$$CH_3\,CH(OH)COO^- + H_2O \rightarrow CH_3COO^- + 2H_2 + CO_2 \quad \Delta G_r^o = -8.79\ kJ/mol \quad (1)$$

$$CH_3CH(OH)COO^- + CH_3COO^- + H^+ \rightarrow CH_3(CH_2)_2COO^- + H_2O + CO_2 \quad \Delta G_r^o = -57.52\ kJ/mol \quad (2)$$

$$CH_3CH(OH)COO^- + CH_3(CH_2)_2COO^- + H^+ \rightarrow CH_3(CH_2)_4COO^- + H_2O + CO_2 \quad \Delta G_r^o = -57.7\ kJ/mol \quad (3)$$

$$CH_3CH_2OH + H_2O \rightarrow CH_3COO^- + H^+ + 2H_2 \quad \Delta G_r^o = 10.5\ kJ/mol \quad (4)$$

$$6\,CH_3CH_2OH + 4\,CH_3COO^- \rightarrow 5\,CH_3(CH_2)_2COO^- + H^+ + 2H_2 + 4\,H_2O \quad \Delta G_r^o = -182.50\ kJ/mol \quad (5)$$

$$6\,CH_3CH_2OH + 5\,CH_3(CH_2)_2COO^- \rightarrow CH_3COO^- + 5\,CH_3(CH_2)_4COO^- + H^+ + 2\,H_2 + 4\,H_2O \quad \Delta G_r^o = -183.50\ kJ/mol \quad (6)$$

The FAB pathway is widely present in micro-organisms, as the fatty acids produced through the FAB pathway are mainly used to synthesize many building blocks of cell membranes, such as sterols, sphingolipids, and phospholipids [29]. Unlike the RBO pathway, in the FAB process, acetyl CoA first consumes ATP through the action of acetyl CoA carboxylase to convert to malonyl CoA, and then forms malonyl ACP [16]. The first cycle of FAB begins with the formation of acetyl ACP by acetyl CoA and malonyl ACP catalyzed by ketoacyl synthase. In this reaction, it is catalyzed by three enzymes, including ketoacyl ACP reductase, hydroxyacyl ACP dehydratase, and enoyl ACP reductase [12,40]. Subsequently, acetoacetyl ACP is reduced to butyryl ACP, which enters the next FAB cycle to generate hexyl ACP. The generated acyl ACP is hydrolyzed with the assistance of thioesterase to produce the corresponding fatty acids [40]. Because acetyl CoA is further converted into malonyl ACP in the FAB pathway, and ATP is consumed during the production of malonyl ACP, the FAB pathway is theoretically more difficult to achieve than the RBO pathway.

## 3. Factors Affecting the AD of Organic Waste to Produce Intermediate-Chain Carboxylic Acids

Factors influencing organic waste include temperature, pH value, electron donor/electron acceptor ratio, fermentation method, and other factors such as hydraulic retention time (HRT), organic loading rate (OLR), and hydrogen partial pressure. We reviewed the impact of each factor on MCCA in the past five years using the Web of Science database. Specifically, there were 81 studies that focused on substrates, 87 on temperature, 71 on pH values, 32 on electron donors and acceptors, and 11 on OLR, and the least studied were HRT and hydrogen partial pressure with only five and six studies, respectively. The results are shown in Table 1. The following section will examine the impact of each relevant factor on the production of MCCA through the process of AD.

**Table 1.** Analysis of relevant factors affecting MCCA in the past five years.

| Related Factors Affecting MCCA | Substrate Type | Temperature | pH | OLR | HRT | ED/EA | Hydrogen Partial Pressure |
|---|---|---|---|---|---|---|---|
| Number | 81 | 87 | 71 | 11 | 5 | 32 | 6 |

### 3.1. Temperature

The temperature has a significant impact on the growth and metabolic processes of micro-organisms, and an appropriate temperature range can promote the growth of micro-organisms and the synthesis of metabolites [41]. Some literature has shown that the growth rate of micro-organisms and the synthesis of metabolites increase with the increasing temperature within a certain range, but beyond this range, temperature can become a limiting factor [42,43]. For example, Zhu et al. (2017) studied the effect of different temperatures (20, 30, 40, and 50 °C) on the yield of caproic acid in *Ruminococcaceae strain CPB6*. The results showed that the growth and caproic acid production of *Ruminococcaceae strain CPB6* were ideal at temperatures of 30 °C and 40 °C, but were limited at 20 °C and 50 °C [43]. Wu et al. (2020) increased the yield of MCCA by 61.17 ± 2.90% by increasing

the reaction temperature from 35 °C to 40 °C. As the temperature continued to rise or fall, the production performance of MCCA deteriorated. In the AD reactor, the environment of most micro-organisms is between 30 and 40 °C, such as *Clostridium kluyveri*, *Eubacterium pyruvativorans*, *Eubacterium limosum*, *Megasphaera elsdenii*, and *Clostridium* sp. *BS-1* [42]. However, there are exceptions, such as Sakarika et al. (2023)'s study on the production of caproic acid under thermophilic conditions, where using grass juice to produce caproic acid at 50 °C is advantageous, and the presence of xylose is beneficial for the presence of Caprociproducts [44]. Compared with medium temperatures, high temperatures can allow for a higher substrate hydrolysis rate, which can solve the problem of hydrolysis rate limitation [45]. At the same time, high temperatures can also solve the problem of pathogens in the substrate [41].

### 3.2. pH

pH value has a significant impact on the growth and metabolic processes of micro-organisms [46]. The appropriate pH range can promote the growth of micro-organisms and the synthesis of metabolites, while also inhibiting methanogens to a certain extent [42]. For example, Zhu et al. (2017) found that the pure strain of *Ruminococcaceae strain CPB6* can synthesize hexanoic acid in the pH range of 5.0–6.5. When the pH is between 5.5 and 6.0, the synthesis efficiency of hexanoic acid is the highest. However, under neutral conditions (pH: 7.0–8.0), the bacterial utilization efficiency of lactic acid significantly decreases, and the synthesis of hexanoic acid cannot be detected [43]. Candy et al. (2020) used lactic acid as an electron donor, and the impact of open lactic acid fermentation on pH value was studied. It was found that, when the pH value was greater than 6, the fermentation path moved to propionic acid, while, when the pH value was less than 6, the fermentation product was close to the mixture of butyric acid, valeric acid, and caproic acid. The research results indicate that, under low pH conditions, lactic acid is not conducive to the fermentation of propionic acid, and its products are converted into caproic acid or carboxylic acids with longer carbon chains [47]. Yu et al. (2019) studied the activity of *Kluyveri* under different pH values (5.5, 7.5, 8.5, and 9.5) with ethanol as the electron donor. At a pH of 7.5, *Kluyveri* has the highest activity. At pH 5.5, 8.5, and 9.5, *Kluyveri* was inhibited to varying degrees. *Kluyveri* bacteria have been proven to be a key colony for CE [48]. Wu et al. (2020) studied the concentration changes of MCCA under pH conditions of 5.2, 5.4, 5.6, 5.8, 6, and 6.5, and found that MCCA concentration was highest at pH 5.4. Compared to pH 6.5, the concentration of MCCA increased by approximately 10.76%. However, at other pH values, the concentration of MCCA was also inhibited differently. This result indicates that, at an appropriate pH value, not only does it increase the concentration of MCCA, but it also inhibits the activity of methanogens, avoiding the reduction of inhibitors such as 2-bromoethanesulfonic acid [42]. It should be noted that, when the concentration of undissociated carboxylic acids is 0.2–0.3 $g \cdot L^{-1}$, it will inhibit the production of MCCA [16]. Therefore, it is important to set the pH value reasonably for AD-producing MCCA. The pH in AD is generally neutral, ranging from 5 to 6, which not only provides a suitable environment for micro-organisms but also inhibits methane production and promotes MCCA production.

### 3.3. Ratio of ED/EA

ED/EA is an important influencing factor in MCCA production [31]. When ethanol is used as the ED, the low proportion of ED/EA results in the lack of electron donors in the system, which is insufficient to produce MCCA [49]. However, the high proportion of ED/EA will produce toxic effects on cells that produce CE, leading to a decrease in the production of MCCA [48]. For example, Yu et al. (2019) added a pure strain of *Clostridium kluyveri* to study the ratio of ethanol to SCCA (0:1, 1:1, 2:1, 3:1, 4:1, and 5:1). As the ethanol ratio increased from 1:1 to 4:1, the yield of caproic acid also increased from 1.0 to 12.1 $g \cdot L^{-1}$. When the ratio was 5:1, ethanol severely inhibited the growth of *Clostridium kluyveri* bacteria, resulting in a decrease in hexanoic acid production to 2.5 $g \cdot L^{-1}$ [48].

When lactic acid is used as the electron donor, a low proportion of ED/EA will lead to the lack of electron donors in the system, leading to an incomplete CE process, resulting in low MCCA production [38]. However, a high proportion of ED/EA will not have toxic effects on micro-organisms but will activate the acrylic acid pathway, ultimately reducing the production of MCCA [50]. Some literature suggests that the propionic acid pathway is a factor affecting MCCA. As Tang et al. (2022) studied, when the ratio of lactic acid to acetic acid was 3, the highest content of caproic acid was 11.02 mmol/L. When the concentration of ED/EA further increased, it was found that high concentrations of lactic acid activated the acrylic acid pathway, reducing the production of caproic acid [37]. Recent studies have also shown that when ethanol, lactic acid, and SCCA coexist, the optimal ratio is 2:1:1 [51]. Overall, the appropriate types and concentrations of EA and ED contribute to the production of MCCA.

*3.4. Substrate Type*

In the system of utilizing organic waste AD to produce MCCA, fermentation is divided into single fermentation mode and co-fermentation mode based on the amount of substrate [52]. Single fermentation is a separate raw material, such as alcohol wastewater, lactic acid whey, food waste, and sludge [12]. They can utilize their fermentation to produce corresponding ED or EA to produce MCCA. For example, Xu et al. (2018) used a two-stage fermentation process, first using lactic acid whey as raw material to produce lactic acid. The yield of lactic acid during fermentation reached $1.54 \text{ g} \cdot \text{L}^{-1} \text{ h}^{-1}$. Subsequently, hexanoic acid synthesis was carried out at a temperature of 30 °C and pH control at 5.0, and the highest yield of hexanoic acid was $1.68 \text{ g} \cdot \text{L}^{-1} \text{ d}^{-1}$ at an HRT of 2.1 days. Under the condition of no additional electron donor, only relying on the lactic acid fermentation of the organic matter itself, two stages of fermentation were used to optimize their respective conditions, and high-value-added medium-chain fatty acids were efficiently synthesized [17]. Zhu et al. (2022) used liquor brewing wastewater to ferment, first implementing the lactic-acid-driven CE process in a medium-sized laboratory, expanding the lactic-acid-driven CE process from 2.5 L to 500 L, and achieving a yield of $14.5 \pm 0.6 \text{ g} \cdot \text{L}^{-1}$ hexanoic acid in 96 h [53]. Zhang et al. (2023) studied the situation where food waste does not add external electrons. Under a semi-continuous reaction operation, after 120 days, the highest production of total MCCA was $29.88 \text{ g COD} \cdot \text{L}^{-1}$, and the production of caproic acid was $28.19 \text{ g COD} \cdot \text{L}^{-1}$ [29].

Co-fermentation refers to the fermentation of two or more raw materials, which not only enables the system to achieve better C/N, but also increases the richness and diversity of CE micro-organisms, inhibits competition, and strengthens the CE pathway to promote the production of MCCA [19,52,54]. For example, Yin et al. (2022) studied the co-fermentation of sewage sludge and large algae under different mixing ratios for the production of MCCA. The results showed that, in the co-fermentation group with a sludge/macroalgae ratio of 4:6, the highest concentration of MCCA was 112.7 mmol C/L. Macroalgae provided abundant readily available organic matter for AD, while sewage sludge provided the alkaline buffering capacity for AD, resulting in higher MCCA production in the co-fermentation group compared to single fermentation [19]. An appropriate C/N will increase the production of MCCA. Yin et al. (2022) determined the C/N ratio of the system by adjusting the ratio of substrates (antibiotic fermentation residues and ginkgo biloba leaves). When the C/N ratio of the AD system was 50, MCCA production was the highest, at 133.14 mmol C/L. An appropriate C/N ratio was found to optimize substrate conditions, promote leaf hydrolysis, stimulate the enrichment of CE micro-organisms, and strengthen the RBO and FAB pathways of CE [52]. Yin et al. (2022) reported that the co-fermentation of sludge and lignocellulosic biomass can significantly promote the production of MCCA, and the yield of caproate increased by 1.03~41.73% compared to the single fermentation of sludge [54].

*3.5. Other Factors*

In addition to the above influencing conditions, there are also some factors such as HRT and OLR. An appropriate HRT can not only control the structure of micro-organisms in mixed fermentation and reduce unfavorable micro-organisms for synthesizing caproic acid in the system, but also reduce the concentration of undissociated caproic acid, thereby reducing its toxicity to micro-organisms [10]. However, HRT and OLR are related to the fermentation of raw materials and reactors, as hydrolysis is limited in speed without prior pretreatment (hydrolysis) of the fermentation of raw materials, and stirring or pumping is impractical [9]. Therefore, a longer residence time is required for hydrolysis and dissolution. Only after pre-treatment can a decrease in HRT and an increase in OLR lead to an increase in CE [55]. For example, Fernando Foncillas et al. (2021) found that, through the co-fermentation of food waste and sludge, the concentration of caproic acid increased by 44% when HRT decreased from 4 days to 2 days [55]. Zhang et al. (2022) added acidified feces and ethanol to the CE reactor and studied an OLR of 7.0, 13.5, and 18.5 g COD/L/d. They found that the system can stably produce MCCA with a maximum concentration of 27.6 g COD·L$^{-1}$ when the OLR is 13.5 g COD/L/d. However, at the same time, an increase in OLR will increase the concentration of ammonia in the system, thereby reducing the production of MCCA [14]. HRT and OLR are also related to temperature and pH, and further research is needed for different systems [9].

Hydrogen partial pressure is a key parameter limiting the competitive process [10]. During the CE process, the oxidation of ethanol to acetic acid produces $H_2$, which can also reduce acetic acid to ethanol. Therefore, the content of $H_2$ is an important factor affecting AD [56]. The partial pressure of $H_2$ has a significant impact on the thermodynamic feasibility of various microbial pathways. Under very low $H_2$ partial pressure (approximately 1Pa), the oxidation of carboxylates is feasible, while relatively high pressure (approximately 10 kpa $H_2$) can avoid excessive ethanol oxidation [56]. Research has shown that low H2 partial pressure (below $5 \times 10^{-2}$ atmospheres) may result in a high butyrate concentration instead of caproate concentration [57]. At present, the mechanism of hydrogen partial pressure on MCCA generation is still incomplete. Therefore, it is necessary to further explore the impact of specific hydrogen partial pressure on caproic acid production and microbial activity in the future.

Further research is necessary to understand the mechanisms by which different factors influence MCCA production. It is crucial to understand the key driving factors that optimize MCCA production by controlling the main factors influencing the process of microbial conversion of organic substrates to MCCA. However, the factors determining MCCA can also be used to simulate the process of anaerobic digestion acid production through mathematical modeling. Coelho et al. (2020) developed several kinetic models to predict and simulate MCCA generation, using residual glycerol and dairy wastewater substrates to evaluate the feasibility of MCCA production [58,59]—for example, cone models, Fitzhugh, and first-level models. The model equation is shown in Equations (7)–(9). Fei Long et al. (2022) used machine learning to establish a control model and determined the key operating parameters of the MCCA production process—for example, retention time, temperature, pH, and OLR, where OLR and HRT are identified as key operational parameters that affect the fermentation process [60]. With the continuous development of technology, machine learning and modeling have also provided significant assistance in the generation and understanding of MCCA. How to respond ecologically and effectively to different influencing factors is a question worth considering, as the mutual influence of multiple factors on the generation of MCCA in AD. The impact of various factors on MCCA is shown in Figure 2.

$$\text{First} - \text{order} \quad \text{CA}_t = \text{CA}_f[1 - \exp(-k_{CA}t)] \tag{7}$$

$$\text{Fitzhugh} \quad \text{CA}_t = \text{CA}_f[1 - \exp(-k_{CA}t)^n] \tag{8}$$

$$\text{Cone} \quad CA_t = \frac{CA_f}{1 + (k_{CA}t)^{-n}} \tag{9}$$

where: $CA_t$: concentration over time; $CA_f$: final concentration of CA; $k_{CA}$: first-order CA production rate constant; $t$: digestion time; and $n$: shape constant.

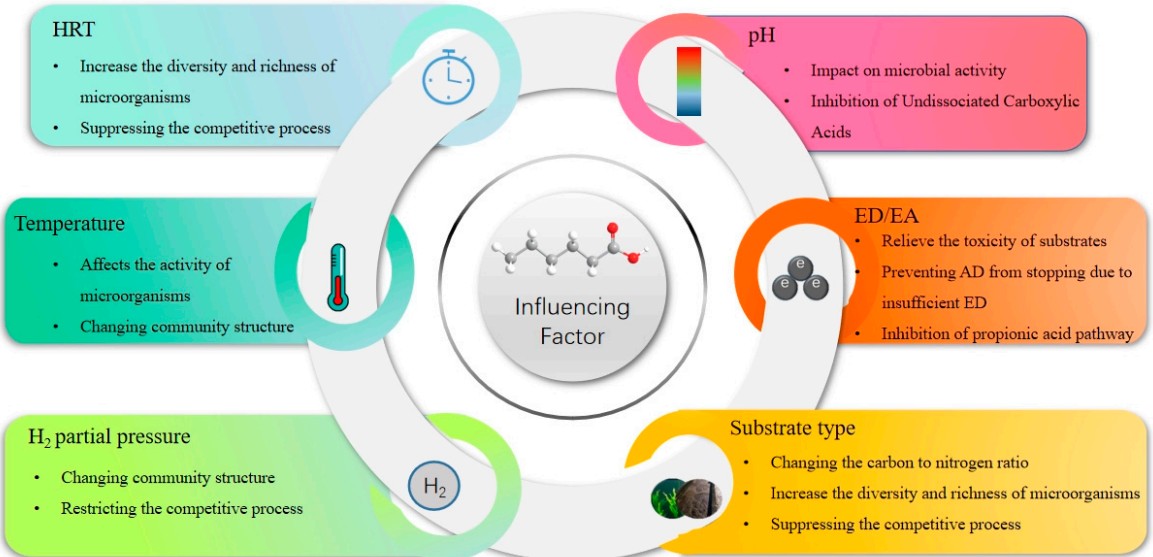

**Figure 2.** Factors affecting MCCA.

## 4. Preparation, Characteristics, and Application of Carbon-Based Materials in AD

### *4.1. Porous Carbon Materials*

#### 4.1.1. Preparation of Porous Carbon Materials

Porous carbon is a carbon-rich product produced by pyrolysis of biomass under the condition of isolating oxygen. Porous carbon materials include biochar and activated carbon. The raw materials used to prepare porous carbon are generally very extensive, such as agricultural waste, animal manure, algae, waste sludge, wood, animal bones, coal, charcoal, etc. [61–65].

There are three methods for preparing biomass—baking, pyrolysis, and hydrothermal carbonization [28]. Baking is a thermal conversion (or pre-treatment) technology that converts biomass raw materials into intermediate solid biofuels (biochar) [66]. Biomass is mainly dried at 200–300 °C in a low-oxygen or inert atmosphere to remove moisture and low-energy volatiles [28]. Compared with raw biomass materials, biochar has higher energy density and calorific value [67]. Pyrolysis occurs at a relatively higher temperature than baking. The process begins with the drying of biomass, where particles are further heated and volatile substances are released from the solid [68]. Volatile compounds can form permanent gases (such as $CO_2$, CO, $CH_4$, and $H_2$) or condensable organic compounds (such as methanol and acetic acid), and the remaining solid is pyrolysis biochar [69]. The four basic categories of biomass pyrolysis are generally distinguished based on the heating rate and residence time—slow pyrolysis, intermediate pyrolysis, rapid pyrolysis, and flash pyrolysis [70]. The types of pyrolysis are shown in Table 2. Slow pyrolysis of biomass is a thermal decomposition process with a moderate temperature, low heating rate, and long residence time [71]. Usually, the yield of biochar produced by slow pyrolysis is the highest [28]. Intermediate pyrolysis usually depends on the type of raw materials and processing conditions, with a product yield of 10–30% liquid (pyrolysis oil and water), 15–20% gas, and 50–75% biochar [72]. Rapid pyrolysis refers to the rapid heating of biomass under anaerobic conditions at 600–1000 °C to produce pyrolysis steam and biochar. It adopts a fast heating rate (10–10,000 °C $min^{-1}$) and a short residence time (0.5–5 s) to maximize biomass oil production, with biochar as a byproduct [73]. Flash pyrolysis refers

to the maximization of gas production with a higher heating rate and a shorter residence time (<1 s) [28]. The advantage of hydrothermal carbonization is that it does not require the dehydration of raw materials. Compared to pyrolysis, the temperature of hydrothermal carbonization is relatively lower than that of pyrolysis, and the yield of coke obtained is also higher than that of pyrolysis (the carbon yield of hydrothermal carbonization reaches 20–80%). It is a promising alternative thermal conversion technology [74]. During hydrothermal carbonization, wet organic raw materials are generated into liquid, solid, and gas phases at a temperature of 180–250 °C and a self-pressure of 2–10 Mpa. The solid phase contains a large amount of carbon (with a yield of 20–80%) [75].

The preparation methods of activated carbon and biochar are similar, except that activated carbon is activated during preparation to increase the properties of porous carbon materials, such as surface area, porosity, pore volume, and pore size [76]. Activation is usually divided into physical activation and chemical activation. Physical activation is the process of heating carbonized raw materials at high temperatures (800–1100 °C) in a neutral gas atmosphere ($CO_2$, $CO_2$, and $N_2$ mixture) to produce porous activated carbon [77]. Chemical activation refers to the first use of chemical substances to treat carbonized raw materials, such as impregnation. After the mixture is dried, it is then heated within the temperature range of 400–900 °C. Finally, the resulting mixture is repeatedly washed to obtain porous activated carbon. The commonly used chemical substances include KOH, NaOH, $CaCl_2$, $K_2CO_3$, $H_3PO_4$, $H_2SO_4$, $ZnCl_2$, and other activators [78].

Activated carbon generated by chemical methods has a larger specific surface area, but the process requires the use of chemicals, which can have a certain impact on the environment. However, chemical methods require lower activation temperatures, shorter treatment times, and higher carbon efficiency than physical methods, which are favored by researchers [77].

**Table 2.** Types of pyrolysis.

|  | Slow Pyrolysis | Intermediate Pyrolysis | Fast Pyrolysis | Flash Pyrolysis |
|---|---|---|---|---|
| Temperature (°C) | 300–700 | 450–550 | 500–1200 | 900–1300 |
| Heating rate (°C/ s) | 0.1–10 | 1–10 | 10–200 | >1000 |
| Residence Time (s) | 300–7200 | 120–6000 | 0.5–10 | <1 |
| Pressure (Mpa) | 0.1 | 0.1 | 0.1 | 0.1 |
| Reference | [28,71] | [72] | [28,79] | [28] |

4.1.2. Characteristics of Porous Carbon

Usually, the characteristics of porous carbon are manifested as a porous structure, large specific surface area, adsorption, conductivity, redox properties, pH being alkaline, etc. [80]. The preparation process of porous carbon is influenced by the raw materials and pyrolysis parameters (temperature, rate, pyrolysis time, biological plasmid diameter, etc.) [81].

As mentioned above, many kinds of biomass raw materials can be used for the preparation of biochar, and different biomass raw materials contain different substances. General wastes include cellulose, lignin, hemicellulose, protein, etc., as well as other trace elements [68]. Generally, hemicellulose, cellulose, and lignin will gradually decompose at 220–315 °C, 280–400 °C, and 160–900 °C, respectively, while lipids, proteins, and carbohydrates will generally decompose at 259–470 °C, 245–312 °C, and 139–397 °C [28]. It is precisely because of the differences in component composition between each material that the physical properties of biochar prepared from different raw materials vary. Studies have shown that, compared to porous carbon produced from crop residues and wood biomass, porous carbon produced from animal manure and solid waste raw materials has a lower surface area, carbon content, and volatile matter [80]. The properties of porous carbon

are also related to the temperature of pyrolysis. As the pyrolysis temperature increases, biomass raw materials will produce more volatile substances and carbonaceous gases, leading to a decrease in the total amount of acidic functional groups and an increase in alkaline functional groups, resulting in a more alkaline porous carbon. The total nitrogen, hydrogen, and oxygen content of porous carbon will also significantly decrease with the increase in temperature. On the contrary, the porosity, specific surface area micropore volume, total pore volume, and ash content will increase with the increase in pyrolysis temperature [81]. Meanwhile, as the heating rate decreases, the degree of dehydration of carbon materials increases, resulting in a higher carbon content, while an increase in residence time helps to form more pores in the carbon [68]. In addition, if the flow rate of the inert gas is too low, it will take more time to form an inert atmosphere in the furnace, which, in turn, will cause sample ashing due to the presence of oxygen [76].

After pyrolysis, the performance of biochar may not meet practical requirements. To further meet practical requirements, the biochar will be activated [82]. The physical and chemical properties of activated biochar will be improved, such as the specific surface area, porosity, and surface functional groups [83,84]. Through the activation process, the physical and chemical properties of biochar can be enhanced to transform it into activated carbon [43]. The activation techniques are generally divided into physical activation and chemical activation [85,86]. The activation stage affects the properties of activated carbon and has different properties for different activation temperatures, activation times, and types of activators. For example, Masoumi et al. (2020) showed that using algae water compounds as raw materials to prepare activated carbon, under the same conditions, and using $K_2CO_3$ as a chemical agent to prepare activated carbon is more effective than NaOH and KOH [83]. Vakili et al. (2023) used walnut shells as raw materials and used different activation temperatures (700–1000 °C) and activation times (30–120 min) during the same carbonization stage. The results show that the BET-specific surface area of activated carbon prepared at a 60 min activation time and 900 °C activation temperature is the maximum value of 903.91 $m^2/g$ [87].

Usually, activated carbon generated by chemical activation has a larger specific surface area compared to physical activation, but the process requires the use of chemicals, which can have a certain impact on the environment. However, chemical methods are usually more economical because they require lower activation temperatures, shorter treatment times, and higher carbon efficiency, which are favored by researchers [77].

4.1.3. Application of Porous Carbon Materials

Adding carbon-based materials will have a positive impact on AD. Currently, carbon-based materials such as biochar, activated carbon, and graphene are added to AD systems [21,26,27,88]. Carbon-based materials mainly exhibit the ability to shorten the lag period and increase the content of MCCA in AD products [26,27]. The effect of different carbon-based materials on the AD production of MCCA is shown in Table 3. For example, in the study by Liu et al. (2017), for the first time, by adding 20 $g \cdot L^{-1}$ biochar, ethanol as the electron donor, and acetate as the EA, the lag period of AD was shortened by 2.3 times, while the concentration of caproic acid reached 21.1 $g \cdot L^{-1}$ [8]. Wu et al. (2021) found that, by adding biochar in an AD system, the first appearance of MCCA in the group with biochar was two days earlier than in the control group, and the rate of substrate ethanol consumption was also 7 days faster in the biochar group than in the control group. The production of MCCA was also 35% higher than in the control group [11]. Ghysels et al. (2021) found that the dosage of adding activated carbon and biochar was 10 $g \cdot L^{-1}$, and the yield of caproic acid was 9.63 ± 0.58 $g \cdot L^{-1}$ (control), 10.075 ± 0.10 $g \cdot L^{-1}$ (biochar), and 10.61 ± 0.058 $g \cdot L^{-1}$ (activated carbon), respectively [10].

**Table 3.** Effect of carbon-based materials on AD production of medium-chain carboxylic acids.

| Carbon-Based Materials | Character | Substrate | Effect | Reference |
|---|---|---|---|---|
| biochar (20 g·L$^{-1}$) | pH: 9.03, BET: 8.92 m$^2$/g, diameter: 0.5–1.0 mm | ethanol acetate | The concentration of caproic acid was 21.1 g L$^{-1}$, which increased by 1.46 times and shortened the lag period by 21 d. | [21] |
| biochar (20 g·L$^{-1}$) | granularity < 5 mm, BET: 221±25 m$^2$/g, total pore volume: 0.1 ± 3 × 10$^{-3}$ cm$^3$/g | ethanol acetate (3:1) | The concentration of caproic acid is 56 mmol L$^{-1}$, which is 2.8–3.8 times that of the other groups. | [25] |
| biochar (26.4 g·L$^{-1}$) | length dimension < 150 um, PH: 8.95–9.22, BET: 161.5 m$^2$/g | ethanol acetate (3:1) | The concentration of caproic acid increased by 1.15 times compared to the control group. | [88] |
| graphene powder | granularity: 2 mm, length dimension < 2 um | ethanol acetate | Extension of the lag period. | [88] |
| biochar (20 g·L$^{-1}$) | BET: 221 ± 25 m$^2$/g, total pore volume: 0.1 ± 3 × 10$^{-3}$ cm$^3$/g | ethanol, waste activated sludge | The concentration of caproic acid is 5.7 g COD·L$^{-1}$, which is 35.1 ± 4.9% of the control group. | [27] |
| powdered activated carbon (15 g·L$^{-1}$) | - | ethanol acetate | The concentration of caproic acid is 154.6 ± 8.76 mmol·L$^{-1}$, which is 2.04 times higher than the control. | [16] |
| granular activated carbon (15 g·L$^{-1}$) | alkaline pH, granularity: 0.71–3.15 mm, BET: 875 m$^2$·g$^{-1}$, pore volume: 0.55 cm$^3$·g$^{-1}$, average pore diameter: 2.7 nm | lactic acid, acetic acid | GAC reduces the lag time of CE to 2 days. | [24] |
| activated carbon (10 g·L$^{-1}$) | - | ethanol acetate | The concentration of caproic acid is 10.61 g·L$^{-1}$, which is 1.1 times that of the control group. | [26] |
| biochar (10 g·L$^{-1}$) | - | ethanol acetate | The concentration of caproic acid is 10.08 g·L$^{-1}$, which is 1.05 times that of the control group. | [26] |

According to the different particle sizes of porous carbon materials, they also have different properties, which can affect the efficiency of AD [25,27]. Small particles of porous carbon can be suspended in AD systems, increasing the probability of contact between micro-organisms and porous carbon materials. At the same time, as the particle size of porous carbon materials decreases, their physical properties will also differ. For example, Liu et al. (2020) studied five different particle sizes of biochar (2000–5000, 500–1000, 75–150, 16–25, and <5 um) and found that after 10 days of fermentation, when the concentration of caproate tended to stabilize, the average concentration of biochar with a size less than 5 um in one group was about 56.00 mmol/L, while the average concentration of other groups was 14.60–19.91 mmol/L. It is precise because small particles of biochar are suspended in AD, which increases the probability of contact between biochar and CE micro-organisms, resulting in a cell network structure formed by the bidirectional attachment of dominant strains around biochar particles, thereby enhancing CE [9]. Wu et al. (2021) also conducted related experiments and found that as the particle size of biochar decreased from 2000–5000 to 75–150 um, the surface area and total pore volume of biochar correspondingly increased, resulting in an increase in the cumulative yield of MCCA from 4.8 ± 0.1 g COD·L$^{-1}$ to 5.7 ± 0.3 g COD·L$^{-1}$. The MCCA yield increased by 22.9 ± 4.5% to 35.1 ± 4.9% compared to the control [11].

The number of carbon-based materials added will also have a significant impact on AD. A suitable dosage of carbon-based materials will have a positive effect on CE. For example, Wu et al. (2022) found that when the mass ratio of biochar to the substrate was 1:16, 1:4, 1:1, and 2:1, the yield of caproic acid was 7.11, 7.28, and 8.95 g COD·L$^{-1}$, respectively, which was significantly increased by 12.0%, 14.6%, and 40.9% compared to the control group. When the ratio reached 2:1 (the amount of biochar added was 26.4 g·L$^{-1}$), the highest yield of caproic acid was 13.67 g COD·L$^{-1}$, which was 115% higher than the group without biochar [88]. Ghysels et al. (2021) studied different concentrations of activated carbon and biochar [10]. As the dosage of activated carbon was continuously added, the concentration of caproic acid continued to decrease. When the dosage of activated carbon was 10 g·L$^{-1}$, 20 g·L$^{-1}$, and 30 g·L$^{-1}$, respectively, the yield of caproic acid produced after fermentation was 11.00 ± 0.24 g·L$^{-1}$, 10.45 ± 0.14 g·L$^{-1}$, and 9.91 ± 0.29 g·L$^{-1}$. The biochar was found to be 10 g·L$^{-1}$, 20 g·L$^{-1}$, and 30 g·L$^{-1}$, and it was found that the biochar with 10 g·L$^{-1}$ produced the highest amount of caproic acid. These experiments may be related to the adsorption properties of porous carbon materials [10]. For biochar, the adsorption effect on caproic acid is relatively small. In relevant adsorption experiments, the adsorption amount of 30 g·L$^{-1}$ biochar is only 6%, while the adsorption amount of 30 g·L$^{-1}$ activated carbon is 30%. If hexanoic acid adsorbed on porous carbon can be resolved, it is no different from when the same amount of biochar and activated carbon are added, the production of activated carbon is the highest. Meanwhile, with the addition of porous carbon, the pH value of the system will also increase, which can reduce the toxicity of undissociated carboxylic acids to micro-organisms. For example, Ghysels et al. (2021) found that, with the addition of activated carbon, the pH value of the system also significantly increased (the control group was 6.25, the pH value of 10 g·L$^{-1}$ activated carbon was 6.60, the pH value of 20 g·L$^{-1}$ activated carbon was 6.80, and the pH value of g L$^{-1}$ activated carbon was 6.95) [10]. This is because activated carbon adsorbs undissociated hexanoic acid, promoting an increase in pH value, which can reduce the toxicity of the undissociated hexanoic acid concentration to micro-organisms.

The redox and conductivity of carbon-based materials are closely related to AD. For example, Wu et al. (2021) studied the effect of the redox properties of biochar on AD, using biochar that had undergone chemical treatment to weaken its redox properties. They found that, compared to the control group, the total cumulative increase in MCCA of biochar without weakened redox properties reached 2.05 ± 0.17 g COD·L$^{-1}$, while the total cumulative increase in MCCA of biochar after treatment was 0.38 ± 0.16 g COD·L$^{-1}$ [11]. Wu et al. (2022) also conducted relevant experiments by using chemical reagents to reduce the content of oxygen-containing functional groups (such as -C=O and -COOH) on the surface of biochar [88]. They found that adding biochar that weakens its oxidation-reduction ability increases the delay time. However, using chemical reagents to weaken the oxidation-reduction characteristics of biochar does not affect its conductivity. Compared with the group without adding carbon-based materials, the yield of MCCA is 42.4% higher. This indicates that the redox and conductivity of carbon-based materials may mutually promote the production of MCCA. Excessive activated carbon can adsorb caproic acid, leading to a decrease in production, while biochar is not significant. The addition of carbon-based materials will have a significant impact on AD [10]. For the reuse of recycled carbon-based materials, Ghysels et al. (2021) studied this issue and found that biochar recycling does not affect the second fermentation, but activated carbon recycling affects the second fermentation [10]. Some studies have found that the particle size of carbon-based materials is an important factor affecting AD [9,11], but when the particle size of carbon-based materials is less than 5 um, recycling becomes difficult [9].

*4.2. Graphene*

4.2.1. Preparation of Graphene

Graphene is a new material that tightly stacks carbon atoms connected by sp$^2$ hybridization into a single-layer two-dimensional honeycomb lattice structure [89]. Graphene

is a large family, which is the general name of this kind of material. Generally, graphene is divided into primitive graphene (G), graphite oxide (GO), and reduced graphite oxide (rGO) [89].

The methods for preparing graphene are usually mechanical stripping, redox, SiC epitaxial growth, chemical vapor deposition, liquid phase stripping, and thermal stripping [89]. Chemical vapor deposition (CVD) is a deposition technique that forms thin films on a substrate through chemical reactions of steam and is considered a cheap and effective ecological method for industrial production [63]. Chemical vapor deposition usually uses Cu, Ni, and Co as substrates in an atmosphere of inert carrier gases such as Ar and $H_2$ [63]. In the CVD process, Cu is superior to Ni and Co as metal substrates because the latter has a stronger carbon adsorption capacity than Cu during the growth process [64]. However, in practice, graphene is difficult to manufacture. Generally, reduced graphene and graphite oxide are used as a substitute for graphene. Graphite oxide is the oxidation product of graphene. Its structure and shape are the same as graphene, except that there is an oxygen functional group at the edge of each ring. The oxygen functional group will weaken the performance of the original graphene material, but it can be improved by chemical or thermal reduction methods [90].

### 4.2.2. Characteristics of Graphene

Graphene has good conductivity, heat transfer, optical properties, etc. It is generally used in composite materials, electronics, and medical equipment [90–92].

Usually, graphene with a complete structure has zero band gap, making its electrical properties unregulated. Graphene is inert, extremely difficult to dissolve in organic solvents, and is not easy to composite with other materials [89]. To expand the application of graphene, functionalized graphene through covalent/non-covalent strategies has enhanced its characteristics, such as opening its band gap, improving tuning conductivity, and increasing stability and solubility [93]. Covalent bond modification refers to combining the modified molecules with the functional groups on the surface of graphene through chemical reactions, such as the introduction of modified molecules by a nitro reaction, phosphorylation reaction, and other methods. This method can achieve relatively stable and long-lasting modification effects, but it is necessary to avoid excessive modification leading to a loss of graphene performance. Non-covalent bond modification refers to the use of the π–π stacking effect, van der Waals force, and other interaction forces on the graphene surface to adsorb molecules on the graphene surface, for example, to modify graphene with molecules with a π electronic structure, such as tea extract, aromatic compounds, and so on.

### 4.2.3. Application of Graphene

Adding graphene to the AD system can improve the yield of MCCA to a certain extent [88]. As mentioned above, graphene has strong conductivity and is generally used in electronic devices or medical instruments [90–92]. Related studies have shown that conductivity can enhance CE, and carbon-based materials can construct non-biological conductive channels to reduce energy consumption in the cell synthesis of extracellular conductive pili and c-type cytochrome, thereby accelerating inter-species electron transfer among micro-organisms in co-culture systems [68]. For example, Wu et al. (2022) demonstrated that graphene has an excellent electron transfer ability, with the addition of a small amount of graphene (0.83 g·$L^{-1}$) exhibiting a conductivity of 31.34 μ S/cm, and a higher conductivity of biochar (13.20 g·$L^{-1}$) (42.42 μ S/cm). This significant difference did not lead to an increase in MCCA production. On the contrary, adding the same dose of graphene and biochar resulted in the same production of MCCA, while the lag period was prolonged in the group with graphene added. This indicates that there may be a threshold for conductivity, and once it exceeds a certain value, conductivity may not play a role in AD, and, more importantly, the porous structure and redox properties of biochar play a

role [88]. However, there are currently no relevant reports on the impact of the conductivity threshold on AD.

## 5. Mechanism Analysis of Carbon-Based Materials in AD

### 5.1. Optimization of Microbial Community Structure Using Carbon-Based Materials

Adding carbon-based materials can greatly alter the community structure of AD micro-organisms. Because most carbon-based materials such as biochar and activated carbon have characteristics such as a large specific surface area and large total pore volume, which are more conducive to the movement and attachment of related micro-organisms in their external and internal pores, they have been widely used for cell fixation and microbial growth [21,27]. The presence of porous structures of carbon-based materials in AD systems can serve as carriers for micro-organisms, providing growth sites for anaerobic bacterial communities (such as *Firmicutes*, etc.) [27]. For example, Liu et al. (2017) found that in AD systems with biochar added, micro-organisms adhered to the surface of biochar, and a dense cell and extracellular polymer substances (EPS)-composed aggregate were found around the biochar [21]. This good structure can aggregate micro-organisms with the same function and establish stable relationships with corresponding nutritional partners, thereby improving the metabolic rate of micro-organisms [94]. Wu et al. (2021) found that fine biochar (75–150 μm) has a larger specific surface area and total pore volume than coarse biochar (2000–5000 μm), which is more conducive to the movement and attachment of micro-organisms in their internal and external voids. Compared with the group without the addition of biochar, the colony of *Firmicutes* increased from 28.6% to 60.7%, and the genus *C. kluyueri* increased from $9.1 \times 10^{-2}$ to 27.4% [27]. This contradicts the study by Liu et al. (2020), who found that the pore structure in biochar may not be the main factor contributing to microbial aggregation, as biochar with a surface less than 5 mm was found to be smooth and lacked pores. However, it was also found that the aggregation of CE-related micro-organisms (*Methanosaeta*, *Metanoallis*, *Defluvititoge*, and *Metanobacterium*) is attributed to the high specific surface area, carbon content, true density, and Kp content of small-particle biochar [25].

Biochar and activated carbon have adsorption, conductivity, and redox properties, promoting the microbial achievement of DIET [25,88]. In AD systems, there are two main forms of DIET action: One is the use of the cytochrome and conductive pili of cells to directly contact cells to complete the electron transfer; another approach is the use of carbon or iron-based materials as intermediates to construct nonbiological channels, reducing the energy consumption of cytochrome, conductive pili, and other direct electron transfer processes [95]. For example, Liu et al. (2020) found that adding small particles of biochar increases the probability of contact between biochar particles and CE micro-organisms in the suspended state of the small particles. At the same time, the EPS secreted by cells leads to microbial aggregation, increases microbial contact opportunities, and enhances electron transfer [25]. The quinone and quinone functional groups on the surface of biochar play a redox role, and the redox activity enables biochar to serve as an effective electron shuttle to enhance electron transfer [21]. Wu et al. (2021) demonstrated the degree of electron generation and transfer through electrochemical impedance spectroscopy. The larger the impedance value, the less conducive it is to electron generation and transfer. They found that the impedance values in systems without biochar, without redox properties, and with normal biochar were $48.28 \pm 3.52\ \Omega$, $41.51 \pm 3.46\ \Omega$, and $15.23 \pm 1.24\ \Omega$, respectively [27]. However, the mechanism of DIET in CE is still unclear, and further exploration is needed. According to the above, carbon-based materials may provide conditions for microbial aggregation and the improvement of microbial communities.

### 5.2. Reduce the Toxicity of Substrates and Products

In the AD system, the production of MCCA is inhibited by the substrate (ethanol) and the product of undissociated carboxylic acid [17]. Adding biochar or activated carbon can reduce the impact of ethanol and undissociated carboxylic acids on micro-organisms. The

adsorption ability of carbon-based materials can adsorb cells together, establishing a good structure and improving the cell's resistance to ethanol [21]. For example, Xiang et al. (2022) found that the concentration of ethanol is three times higher than the critical value, but powdered activated carbon at 15–20 g·L$^{-1}$ can still enhance ethanol oxidation and reduce ethanol toxicity [16]. Wu et al. (2021) detected these functional micro-organisms from *Clostridium* spp. as spore-forming bacteria with the ability to tolerate ethanol and MCCA toxicity [27]. The adsorption ability of carbon-based materials can also adsorb undissociated carboxylic acids, thereby reducing the toxic effect of undissociated carboxylic acids on cells. However, reducing the toxicity of undissociated carboxylic acids is also related to the pH of carbon-based materials, which are generally alkaline [96].

Liu et al. (2017) found that in AD systems with the addition of biochar, the concentration of undissolved hexanoic acid in the biochar group was still higher than that in the control group, with a maximum of 0.66 g·L$^{-1}$. However, the biochar group produced more MCCA. The discovery is due to the formation of some locally high pH micro sites around the biochar particles, reducing the concentration of undissociated carboxylates around the biochar and reducing its toxicity to micro-organisms [21]. Xiang et al. (2022) found that the undissociated hexanoic acid exceeded 0.2–0.3 g·L$^{-1}$, but the daily production of the group with the added 15 g·L$^{-1}$ activated carbon was about four times that of the control group [16]. The addition of activated carbon reduced the inhibition of the product on micro-organisms.

Adding biochar and activated carbon can reduce the inhibition of products and substrates in AD systems, ensuring the normal metabolism of micro-organisms and the formation of MCCA.

### 5.3. The Role of Promoting CE

Carbon-based materials such as biochar and activated carbon can enhance the efficiency of CE. With the addition of carbon-based materials, the carbon-based materials stimulate the microbial community to accelerate the oxidation of electron donors and enhance the related genes of RBO and FAB [25,27]. In relevant studies, Liu et al. (2020) found that the concentration of H$_2$ is highest in biochar of less than 5 mm. In the case of methane generation inhibition, H$_2$ can only be produced through the oxidation of ethanol in the chain extension. The addition of biochar accelerates ethanol oxidation, leading to an increase in the production of MCCA [25]. Wu et al. (2021) found that, in AD systems, the surface functional groups of biochar promote electron transfer by adding 20 g·L$^{-1}$ biochar. They found that the abundance of enzymes related to ethanol oxidation, such as ethanol dehydrogenase and acetaldehyde dehydrogenase, increased by four and 1.1 times, respectively. Ethanol-dehydrogenase- and aldehyde-dehydrogenase-related enzymes promote ethanol oxidation, generating more acetyl CoA, which can better promote the related RBO cycle and FAB cycle [27]. Xiang et al. (2022) clarified the impact of carbon-based materials on RBO. Using 3 mmol of acetyl CoA and 3 mmol of acetate as substrates, adding 15 g·L$^{-1}$ powdered activated carbon, the study found that the functional genes of alcohol dehydrogenase and acetaldehyde dehydrogenase, which are converted to acetyl CoA in the ethanol oxidation stage, and phosphoacetyltransferase and acetic kinase, which are responsible for the conversion of acetyl CoA to acetic acid, increased by 17–110%. In the RBO stage, during the synthesis of the first cycle of butyric acid, the gene abundance of acetyl-CoA C-acetyltransferase increased by 58.9% compared to the control, and the relative abundance of genes encoding 3-hydroxy butyl-CoA dehydrogenase and 3-hydroxy butyl-CoA dehydratase also increased by 20% and 22%, respectively. The gene abundance of butyryl CoA dehydrogenase significantly increased by 105%. The genes of acetyl CoA acyltransferase-2, 3-hydroxy acyl CoA dehydrogenase, and enoyl CoA hydratase also increase relatively in the cycle of caproate and octanoate formation [16]. Wu et al. (2021) found in their study that the abundance of enzymes related to the FAB cycle pathway was much higher than that of enzymes related to the RBO pathway, and the FAB pathway was

more significantly promoted [27]. Biochar can alter the pathways of RBO and FAB, but FAB is more pronounced. The mechanism of carbon-based materials in AD is shown in Figure 3.

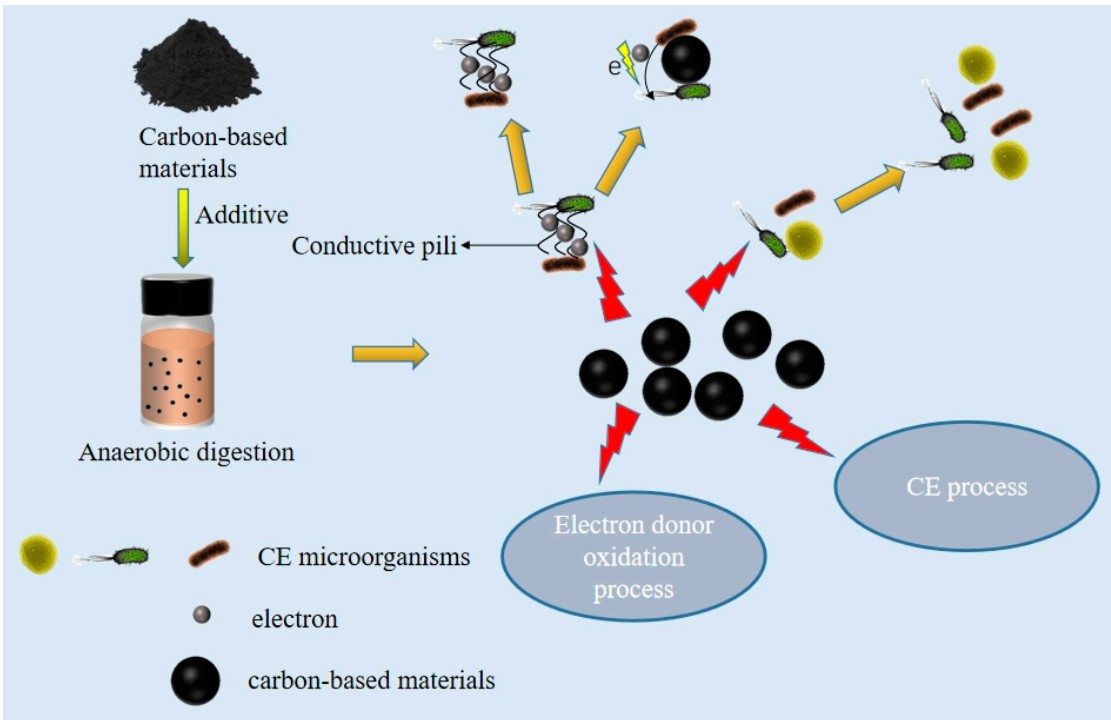

**Figure 3.** Mechanism of carbon-based materials in AD.

## 6. Technical and Economic Assessment

One can realize valuable products (such as biogas, VFA, and MCCA) through organic waste AD, and prepare organic waste into carbon-based materials at the same time. These two methods can not only improve the environment and reduce the pollution of pollutants in the environment, but also help to achieve "carbon neutrality". Adding carbon-based materials to AD can increase the yield of MCCA and shorten the lag time, which not only saves time and costs but also increases the product quantity. The addition of carbon-based materials can also reduce the inhibition of substrates and products, and, in situ, extraction is usually the first choice to reduce product toxicity, while the addition of carbon-based materials can reduce this cost.

The methods for extracting and separating MCCA are adsorption, extraction, and electrodialysis [34]. The downstream separation and extraction of MCCA can account for 40–50% of the total operating cost. As the product increases, the corresponding extraction cost of MCCA will also decrease accordingly [5]. The addition of carbon-based materials can increase AD by 1.05–3 times, and the additional number of carbon-based materials in AD is 10–30 g·L$^{-1}$. For example, Zhu et al. (2022) evaluated the AD technology without adding an electron donor [31]. When considering the cost of fixed assets and labor, it is recommended that the concentration of hexanoate should reach 15 g·L$^{-1}$, if the addition of carbon-based materials can increase the concentration of caproate to about 1.5 times. Taking biochar as an example, the price of carbon-based materials ranges from $0/Mg to $2710/Mg [69], with a median value of $1355/Mg and an additional amount of 20 g·L$^{-1}$. Taking 15 g·L$^{-1}$ hexanoic acid as an example, 22.5 g·L$^{-1}$ hexanoic acid can be produced. If expanded by 1000 times, each ton of biochar can generate an additional 375 tons of hexanoic acid. Assuming the price of caproic acid is approximately $800–$1200 per ton, excluding the price of adding biochar, profits can be made.

### 7. Prospect

This review mainly focuses on the production of MCCA from organic waste in AD. In the first part, CE technology is discussed. Firstly, the RBO pathway has been widely studied in CE technology, but the FBA pathway has been proven to be ubiquitous in micro-organisms. Although FBA is longer than RBO in the pathway, FAB may be more active than RBO in the mixed culture. Therefore, more research is needed to demonstrate the correlation between carbon-based materials and the FAB and RBO pathways. The influencing factors on CE include temperature, pH value, ED/EA ratio, fermentation method, and other factors. These factors can affect the activity of micro-organisms and the production of MCCA, requiring different living environments for different micro-organisms. Research has shown that a mixed culture is more stable than a pure culture, which also provides some insights into the AD of organic waste to produce MCCA. The correlation between different micro-organisms is still unclear, and a deeper understanding of the relationship between bioreactor performance, raw material composition, operating conditions, and microbial community succession should be needed. For traditional AD-producing MCCA, it will be inhibited by substrates and undissociated carboxylic acids, reducing the production of MCCA. Adding carbon-based materials can reduce the inhibition of substrates and undissociated carboxylic acids. By adding carbon-based materials at the same time, the products of AD will increase and the lag time will also decrease. The main reason is that biochar and activated carbon have good physical and chemical properties, which increase the production of MCCA by stabilizing the community structure, reducing the toxicity of substrates and products, enhancing the electron transfer of micro-organisms, and increasing enzyme activity related to CE. In the selection of carbon-based materials, biochar is superior to other carbon-based materials due to its wide source of raw materials, low price, and good performance in AD. However, the promotion of AD by DIET using carbon-based materials is still unclear, and the role of micro-organisms in CE is not sufficiently in-depth. The mechanism of action of carbon-based materials on micro-organisms should be further explored in the future. Researchers should also consider how carbon-based materials can be combined with online separation and extraction techniques to reduce the cost of extracting MCCA. In the future, carbon-based materials will still be a key focus in the AD production of MCCA.

### 8. Conclusions

This review reviewed the chain extension technology of AD, and the factors affecting the chain extension, including temperature, pH value, the ratio of EA to ED, the mode of fermentation, and the impact of other factors (the impact of organic loading rate, hydraulic retention time, and $H_2$). AD can significantly increase the yield of MCCA and shorten the lag period by adding carbon-based materials. Carbon-based materials can come from organic waste, which can also be fermented into MCCA. Carbon-based materials are mainly manifested in two aspects of AD. One is to construct nonbiological channels for micro-organisms through the oxidation reduction and conductivity of carbon-based materials, thereby improving the electronic efficiency of AD. Secondly, through the adsorption and porosity of carbon-based materials, micro-organisms related to CE can be aggregated to establish a good structure, which can better promote the electron transfer of micro-organisms and resist environmental damage. Finally, MCCA can be separated through separation technology. Considering the economic costs, the addition of carbon-based materials can bring benefits to the AD of MCCA, while also improving the separation technology and the application of carbon-based materials. Efficient, low-cost, and large-scale separation methods should be used to improve the availability of caproic acid. I hope this work can provide a theoretical reference for the development of chain carboxylic acid technology in AD production in the future.

**Author Contributions:** Conceptualization, L.J., Y.L. and Q.B.; writing—original draft preparation, L.J., Y.L. and C.C.; writing—review and editing, M.L., Y.X. and Q.B.; project administration, M.L. and Y.X. All authors have read and agreed to the published version of the manuscript.

**Funding:** The authors acknowledge the financial support from the Project Jiangsu Agriculture Science and Technology Innovation Fund (CX(21)2015), Central Public-Interest Scientific Institution Basal Research Fund (GYZX210514), Natural Science Foundation of Jiangsu Province (BK20201242), and China Scholarship Council (Grant No. 202008320123).

**Institutional Review Board Statement:** Not applicable.

**Informed Consent Statement:** Not applicable.

**Data Availability Statement:** Not applicable.

**Conflicts of Interest:** The authors declare no conflict of interest.

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
