# Peer review of "Preparation and Application of Carbon-Based Materials in the Production of Medium-Chain Carboxylic Acids by Anaerobic Digestion: A Review"

_fermentation, doi:10.3390/fermentation9070586_

Round 1

Reviewer 1 Report

Dear Jiao et al.:

With reference to the review entitled "Preparation and application of carbon-based materials in the production of medium chain carboxylic acids by anaerobic digestion: A review", it seems to me a robust and sustainable manuscript, in which I have small observations. Therefore, I recommend its acceptance with minor revisions.

Comment #01: There is a typo on line 128

Comment #02: What is the Gibbs free energy of reaction 5?

Comment #03:Table 2, the use of the word source, means where the material or waste comes from. Please change to reference

Comment #04: I feel the lack of a graph with the number of manuscripts in the last 5 years, for each parameter analyzed.

Reviewer 2 Report

The review work is very well presented and discussed, there is also a good review of the bibliography, and the subject is approached from different perspectives. I would only like to suggest that the authors include a section on the mathematical modeling of the process since an important group conducts their research on this approach.

No comments
